# CROSS-MODAL RAG: SUB-DIMENSIONAL TEXT-TO-IMAGE RETRIEVAL-AUGMENTED GENERATION

## ABSTRACT

Text-to-image generation increasingly demands access to domain-specific, fine-grained, and rapidly evolving knowledge that pretrained models cannot fully capture, necessitating the integration of retrieval methods. Existing Retrieval-Augmented Generation (RAG) methods attempt to address this by retrieving globally relevant images, but they fail when no single image contains all desired elements from a complex user query. We propose Cross-modal RAG, a novel framework that decomposes both queries and images into sub-dimensional components, enabling subquery-aware retrieval and generation. Our method introduces a hybrid retrieval strategy—combining a sub-dimensional sparse retriever with a dense retriever—to identify a Pareto-optimal set of images, each contributing complementary aspects of the query. During generation, a multimodal large language model is guided to selectively condition on relevant visual features aligned to specific subqueries, ensuring subquery-aware image synthesis. Extensive experiments on MS-COCO, Flickr30K, WikiArt, CUB, and ImageNet-LT demonstrate that Cross-modal RAG significantly outperforms existing baselines in the retrieval and further contributes to generation quality, while maintaining high efficiency.

## 1 INTRODUCTION

Text-to-Image Generation (T2I-G) has witnessed rapid progress in recent years, driven by advances in diffusion models (Rombach et al., 2022; Saharia et al., 2022; Nichol et al., 2021) and multimodal large language models (MLLMs) (OpenAI, 2025; Google, 2025; Jin et al., 2023), enabling the synthesis of increasingly realistic and diverse images from natural language descriptions. However, in many real-world applications, domain-specific image generation requires knowledge that is not readily encoded within pre-trained image generators, especially when such information is highly long-tailed, fast-updated, and proprietary. To address this limitation, Retrieval-Augmented Generation (RAG) has emerged as a promising paradigm by incorporating an external image database to supply factual reference during generation (Zheng et al., 2025; Zhao et al., 2024). Notable RAG-based image generation approaches such as Re-Imagen (Chen et al., 2022), RDM (Blattmann et al., 2022), and KNN-Diffusion (Sheynin et al., 2022) integrate retrieved images with diffusion models to

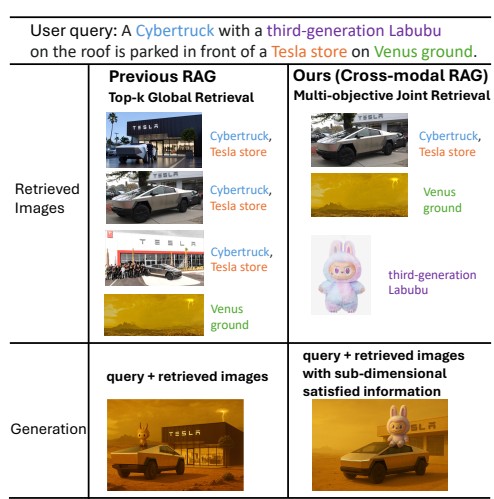

Figure 1: Visualization of retrieval and generation in Cross-modal RAG (ours) versus previous RAG.

improve output fidelity. However, these existing RAG methods typically rely on off-the-shelf retrievers (e.g., those based on CLIP (Radford et al., 2021)) which compute global image-text similarities and retrieve whole images based on the full user query. This coarse-grained retrieval strategy often fails in complex scenarios where the query involves multiple fine-grained entities or attributes (Varma et al., 2023) – especially when no single image contains all required components in the query. In practice, it is extremely common that single images in the retrieval database only satisfy a subset of the query. As in Fig. 1, no single image in the retrieval database perfectly covers all four aspects in

the query; instead, each covers different subsets of the query. Existing RAG methods often retrieve top-k images based on the entire query, so images that redundantly contain most aspects of the query tend to be retrieved (e.g., three images that all include "Cybertruck" and "Tesla store"), while some aspects may be underweighted (e.g., "Venus ground") or even missed (e.g., "third-generation Labubu"), leading to the distortion in the missed aspects. Also, during generation, existing RAG has not been precisely instructed about which aspects of each image should be leveraged, resulting in the superfluous lightning in the generated image by previous RAG.

Therefore, instead of being restricted to considering whether each whole image is related to the entirety of the query, it is desired to advance to pinpointing which aspects (i.e., sub-dimensions) of which images can address which aspects of the query for image generation. Such desire is boiled down to several open questions. *First, how to precisely gauge which part of the queries match which aspects of each image?* Existing global embedding methods, like CLIP, do not naturally support sub-dimensional alignment (Radford et al., 2021), and current fine-grained vision-language matching is limited to region-level object patterns (Varma et al., 2023; Zhong et al., 2022), which are computationally expensive and error-prone. *Second, how to retrieve the smallest number of images that cover all necessary information?* It is desired to retrieve an optimal set of images such that each covers different aspects of the query while avoiding redundancy in order to maximize the amount of relevant information under a limited context window size. *Third, how to precisely inform the image generator of what aspect of each image to refer to when generating?* Existing image generators typically can take in the input query or images, but here it requires adding fine-grained instructions about how to leverage relevant aspects of images when generating, which is not well explored.

To address these open problems, we propose *Cross-modal Sub-dimensional Retrieval Augmented Generation (Cross-modal RAG)*, a novel text-to-image retrieval-augmented framework that can identify, retrieve, and leverage image sub-dimensions to satisfy different query aspects:

- To decompose and identify key image sub-dimensions, we decompose the user query into sub-queries and candidate images into sub-dimensional representations with respect to the subqueries, enabling accurate subquery-level alignment.

- To retrieve comprehensive and complementary image sub-dimensions, we formulate the retrieval goal as a multi-objective optimization problem and introduce an efficient hybrid retrieval strategy – combining a lightweight sub-dimensional sparse retriever with a sub-dimensional dense retriever – to retrieve a set of Pareto-optimal images that collectively cover all subqueries in the query as in Fig. 1 (right).

- To effectively instruct image generators with the retrieved image sub-dimensions, we present a model-agnostic and subquery-aware generation with MLLMs, which explicitly preserves and composes the subquery-aligned components from the retrieved images into a coherent final image. For instance, our method only preserves the "Venus ground" in the final image, while previous RAG can also preserve the irrelevant lightning in Fig. 1.

Extensive experiments demonstrate that Cross-modal RAG achieves state-of-the-art performance in the text-to-image retrieval and also benefits generation tasks across multiple fine-grained, domain-specific, and long-tailed image benchmarks, while maintaining excellent computational efficiency.

## 2 RELATED WORK

### 2.1 TEXT-TO-IMAGE GENERATION

Text-to-Image Generation (T2I-G) has made significant strides, evolving through methodologies such as Generative Adversarial Networks (GANs) (Goodfellow et al., 2014; Brock et al., 2018), auto-regressive models (Van Den Oord et al., 2016; Ramesh et al., 2021), and diffusion models (Ho et al., 2020; Nichol & Dhariwal, 2021). Recent breakthroughs in diffusion models and multimodal large language models (MLLMs), driven by scaling laws (Kaplan et al., 2020), have significantly advanced the capabilities of T2I-G. Notable examples include the DALL-E series (Ramesh et al., 2021; Betker et al., 2023), the Imagen series (Saharia et al., 2022), and the Stable Diffusion (SD) series (Rombach et al., 2022; Podell et al., 2023; Esser et al., 2024). More recently, image generation functionalities have been integrated directly into advanced MLLMs such as GPT Image (OpenAI, 2025) and Gemini Image Generation (Google, 2025). However, despite these advancements, traditional T2I-G methods often struggle with knowledge-intensive, long-tailed, and fine-grained image-generation

tasks. These scenarios typically require additional context to generate accurate images, necessitating RAG techniques.

## 2.2 TEXT-TO-IMAGE RETRIEVAL

Text-to-Image Retrieval (T2I-R) has become a crucial subtask in supporting fine-grained image understanding and generation. CLIP (Radford et al., 2021) is currently the most widely adopted approach, mapping images and texts into a shared embedding space via contrastive learning. While CLIP excels at coarse-grained alignment, it underperforms in fine-grained text-to-image retrieval, especially in scenes involving multiple objects or nuanced attributes. ViLLA (Varma et al., 2023) explicitly highlights this limitation, demonstrating that CLIP fails to capture detailed correspondences between image regions and textual attributes. SigLIP (Zhai et al., 2023), along with other refinements such as FILIP (Yao et al., 2021) and SLIP (Mu et al., 2022), improves CLIP's contrastive learning framework and achieves superior zero-shot classification performance. However, these methods still rely on global image-text embeddings, which are inadequate for resolving localized visual details required by fine-grained queries.

To address this, recent works on fine-grained text-to-image retrieval (e.g., ViLLA (Varma et al., 2023), RegionCLIP (Zhong et al., 2022)) have adopted region-based approaches that involve cropping image patches for localized alignment. In contrast, our vision-based sub-dimensional dense retriever bypasses the need for explicit cropping. By constructing sub-dimensional vision embeddings directly from the full image, we enable more efficient and effective matching against subqueries.

## 2.3 RETRIEVAL-AUGMENTED GENERATION

Retrieval-Augmented Generation has demonstrated significant progress in improving factuality for both natural language generation (Gao et al., 2023; Zhu et al., 2023) and image generation (Chen et al., 2022; Yasunaga et al., 2022). Most RAG-based approaches for image generation are built upon diffusion models (e.g., Re-Imagen (Chen et al., 2022), RDM (Blattmann et al., 2022), KNN-Diffusion (Sheynin et al., 2022)), but these methods largely overlook fine-grained semantic alignment. FineRAG (Yuan et al., 2025) takes a step toward fine-grained image generation by decomposing the textual input into fine-grained entities; however, it does not incorporate fine-grained decomposition on the visual side. In contrast, our approach performs dual decomposition: (i) the query is parsed into subqueries that capture distinct semantic components, and (ii) the candidate images are decomposed into sub-dimensional vision embeddings aligned with the corresponding subqueries. Furthermore, while existing RAG-based image models typically rely on off-the-shelf retrievers, we introduce a novel retrieval method that combines a sub-dimensional sparse filtering stage with a sub-dimensional dense retriever. Finally, with the recent surge of MLLM-based image generation, we explore how our fine-grained retrieval information can be integrated to guide generation at the sub-dimensional level.

## 3 PROPOSED METHOD

We introduce *Cross-modal RAG*, as shown in Figure 2. The framework consists of four stages: (1) Sub-dimensional sparse retriever based on lexical match on subqueries in Sec. 3.1.2; (2) Sub-dimensional dense retriever based on semantic match on sub-dimensional vision embeddings and textual subquery embeddings in Sec. 3.1.1; (3) Multi-objective joint retrieval to select a set of Pareto-optimal images in Sec. 3.1.3; and (4) Subquery-aware image generation with retrieved images in Sec. 3.2.

The framework of *Cross-modal RAG* focuses on: 1) how to **retrieve** the *optimal* images from the retrieval database given multiple subqueries, and 2) how to **guide** the generator to generate images preserving the *satisfied* subquery features in each retrieved image.

### 3.1 MULTI-OBJECTIVE RETRIEVAL FOR IMAGE GENERATION

### 3.1.1 SUB-DIMENSIONAL DENSE RETRIEVER

Given a user query $Q$ and a candidate image $I_j$, we decompose $Q$ into a set of subqueries $\{q_1, q_2, ..., q_n\}$, where each subquery $q_i$ captures a specific aspect of $Q$, such as object categories or attributes, and we further compute the similarity scores between its normalized sub-dimensional vision embeddings and textual subquery embeddings as follows:

$$S(Q, I_j) = \frac{1}{n} \sum_{i=1}^{n} sim(v_{ji}, t_i). \tag{1}$$

Figure 2: Overview of the *Cross-modal RAG* framework. The framework consists of four stages: (1) Sub-dimensional Sparse Retriever, where images are filtered based on lexical subquery matches; (2) Sub-dimensional Dense Retriever, where candidate images are re-ranked using the mean of pairwise cosine similarities between sub-dimensional vision embeddings and subquery embeddings; (3) Multi-objective Joint Retrieval, where a Pareto-optimal set of images is selected by Eq.6 to collectively cover the subqueries. $\mathcal{P}_f$ is composed of three orange points(●) (solid points are on the line while dashed points are off the line); and (4) Generation, where a MLLM composes a final image by aligning subquery-level visual components from retrieved images.

Here, the similarity score $sim$ is cosine similarity. The similarity scores are aggregated across $n$ subqueries to form an overall similarity metric $S(Q, I_j)$. Images are ranked based on their similarity $S(Q, I_j)$ for the given query, and the top-ranked images are retrieved.

The subquery embeddings $t_i$ with respect to subquery $q_i$ can be computed as:

$$t_i = \Phi_{\text{clip-t}}(\, g(q_i)),  \tag{2}$$

where $g(\cdot)$ denotes the tokenization, and $\Phi_{\text{clip-t}}(\cdot)$ denotes the pre-trained CLIP text encoder.

In terms of the sub-dimensional vision embeddings $v_{ji}$, after the image $I_j$ is fed into a pretrained CLIP vision encoder $(\Phi_{\text{clip-v}}(\cdot))$, a multi-head cross-attention module is introduced, functioning as the vision adapter $f_a$, to compute fine-grained sub-dimensional vision subembeddings:

$$v_{ji} = f_a\left(\Phi_{\text{clip-v}}(I_j), T_i\right).  \tag{3}$$

The vision adapter consists of: 1) A multi-head vision cross-attention layer where the learnable query tokens attend to the vision embeddings extracted from the frozen CLIP visual encoder; 2) A multi-head text cross-attention layer where the output of the vision cross-attention further attends to the subquery embeddings extracted from the frozen CLIP text encoder; 3) An MLP head that maps the attended features to a shared multimodal embedding space, followed by layer normalization. The output $v_{ji}$ represents $I_j$'s ith-dimensional vision embedding corresponding to the core concept of subquery $q_i$, which is decomposed from $Q$ and can be obtained by an off-the-shelf LLM (e.g. GPT-4o mini) using the structured prompt in Appendix A. $T_i$ denotes the embedding of the core concept extracted from $q_i$, representing the object category without attribute modifiers.

The vision adapter $f_a$ is optimized using the Info-NCE loss:

$$\mathcal{L}_{\text{Info-NCE}} = -\log \frac{\sum_{(v_{ji}, t_i) \in \mathcal{P}} \exp\left(\frac{\langle v_{ji}^T, t_i \rangle}{\tau}\right)}{\sum_{(v_{ji}, t_i) \in \mathcal{P}} \exp\left(\frac{\langle v_{ji}^T, t_i \rangle}{\tau}\right) + \sum_{(v'_{ji}, t'_i) \sim \mathcal{N}} \exp\left(\frac{\langle v_{ji}'^T, t'_i \rangle}{\tau}\right)},  \tag{4}$$

where $\mathcal{P}$ is a set of positive pairs with all sub-dimensional vision embeddings and subquery embeddings, $\mathcal{N}$ conversely refers to an associated set of negative pairs. $\tau$ is a temperature parameter.

### 3.1.2 SUB-DIMENSIONAL SPARSE RETRIEVER

**Definition 3.1** (Sub-dimensional Sparse Retriever). For each retrieval candidate image $I_j$, we define a binary satisfaction score for the sub-dimensional sparse retriever:

$$s_i(I_j) = \begin{cases} 1, & \text{if the caption of } I_j \text{ contains } q_i \\ 0, & \text{otherwise} \end{cases}  \tag{5}$$

Hence, each image $I_j$ yields an $n$-dimensional satisfaction vector $[s_1(I_j), \dots, s_n(I_j)]$.

---

**Algorithm 1** MULTI-OBJECTIVE JOINT RETRIEVAL ALGORITHM

---

**Require:** Query $Q$ decomposed into subqueries $\{q_i\}_{i=1}^n$, image retrieval database $\mathcal{D}$, weights $\{\alpha_i\}_{i=1}^n$ with $\alpha_i > 0, \sum_i \alpha_i = 1$, trade-off parameter $\beta$ with $0 < \beta < \beta_{max}$

**Ensure:** The set of Pareto optimal images $\mathcal{P} = \{I_j^*\}$

  1: **for** $I_j \in \mathcal{D}$ **do**
  2:     compute $\mathbf{s}(I_j) = [s_1(I_j), \ldots, s_n(I_j)]$
  3: **end for**
  4: $\widetilde{D} \leftarrow \{I_j \mid \mathbf{s}(I_j) \text{ is not all zeros}\}$
  5: $\mathcal{P} \leftarrow \emptyset$
  6: **for** $\alpha$ in a discretized grid over the simplex **do**
  7:     $\mathcal{P} \leftarrow \mathcal{P} \cup \arg\max_{I_j \in \widetilde{D}} \sum_{i=1}^n \alpha_i s_i(I_j) + \beta \cdot nS(Q, I_j)$
  8: **end for**

---

**Definition 3.2** (Image Dominance). Consider two images $I_a$ and $I_b$ from the retrieval database $\mathcal{D}$, with corresponding subquery satisfaction vectors $\mathbf{s}(I_a) = [s_1(I_a), \ldots, s_n(I_a)]$ and $\mathbf{s}(I_b) = [s_1(I_b), \ldots, s_n(I_b)]$. We say $I_a$ *dominates* $I_b$, denoted $I_a \succ I_b$, if:

**(1)** $s_i(I_a) \geq s_i(I_b), \forall i \in \{1, \ldots, n\}$, and **(2)** $\exists j \in \{1, \ldots, n\}$ s.t. $s_j(I_a) > s_j(I_b)$.

That is, $I_a$ is never worse in any subquery's score and is strictly better in at least one subquery. $I_a \in \mathcal{D}$ is retrieved by the sub-dimensional sparse retriever if there exists no other image $I_b \in \mathcal{D}$ such that $I_b$ dominates $I_a$. Formally, $\nexists I_b \in \mathcal{D}$ s.t. $I_b \succ I_a$.

### 3.1.3 MULTI-OBJECTIVE OPTIMIZATION FORMULATION AND ALGORITHM

Each subquery can be regarded as a distinct objective, giving rise to a multi-objective optimization problem for retrieval. Our primary goal is to select images that collectively maximize text-based subquery satisfaction (sub-dimensional sparse retrieval), while also maximizing fine-grained vision-based similarity (sub-dimensional dense retrieval). Thus, the overall objective is formalized as:

$$I_j^* = \arg\max_{I_j} \sum_{i=1}^n \alpha_i s_i(I_j) + \beta \cdot nS(Q, I_j), \text{ s.t. } \forall\alpha_i : \alpha_i > 0, \sum_{i=1}^n \alpha_i = 1, \beta \in (0, \beta_{\max}), \quad (6)$$

where $\alpha_i$ is the relative importance of each subquery $q_i$ in the sub-dimensional sparse retrieval, and the weight $\beta$ trades off between the sub-dimensional sparse and dense retrieval.

**Definition 3.3** (Pareto Optimal Images). The solution to Eq.6 is called the set of *Pareto optimal images*, such that $I_j^*$ is not dominated by by any other image $I_k \in \mathcal{D}$ in terms of both $\mathbf{s}(I)$ and $S(Q, I)$. Formally,

$$\mathcal{P} = \{I_j^* \in \mathcal{D} \mid \nexists I_k \in \mathcal{D} \text{ s.t. } F(I_k) > F(I_j^*)\}, \quad (7)$$

where $F(I_j) = \sum_{i=1}^n \alpha_i s_i(I_j) + \beta \cdot nS(Q, I_j)$, s.t. $\forall\alpha_i : \alpha_i > 0, \sum_{i=1}^n \alpha_i = 1, \beta \in (0, \beta_{\max})$.

**Definition 3.4** (Pareto Front of the Pareto Optimal Images). $\mathcal{P}$ is sometimes referred to as the *Pareto set* in the decision space (here, the set of images). *Pareto front* $\mathcal{P}_f$ of the Pareto optimal image is the corresponding set of non-dominated tuples in the objective space:

$$\mathcal{P}_f = \{(\mathbf{s}(I_j^*), S(Q, I_j^*)) : I_j^* \in \mathcal{P}\}. \quad (8)$$

Therefore, the Pareto optimal images in $\mathcal{P}$ represent the "best trade-offs" across all subqueries, since no single image in $\mathcal{D}$ can strictly improve the Pareto front $\mathcal{P}_f$ on every subquery dimension.

We propose the multi-objective joint retrieval algorithm in Algorithm 1. If an image is Pareto optimal, there exists at least one choice of $\{\alpha_i\}$ for which it can maximize $\sum_{i=1}^n \alpha_i s_i(I_j)$. In particular, if multiple images share the *same* subquery satisfaction vector $\mathbf{s}(I_j)$, we can use the sub-dimensional dense retriever to further distinguish among images.

**Theorem 3.1** (Retrieval Efficiency). *Let $N$ be the total number of images in $D$, $\widetilde{N} \ll N$ be the number of images in $\widetilde{D}$, $K$ is a grid of $\alpha$-values and $n$ be the number of subqueries. Also let $T_{clip}$ represent the cost of processing a single image with the CLIP vision encoder, and $T_{adaptor}$ represent the cost of the adaptor. The time complexity of Algorithm 1 is $\mathcal{O}(N) + \mathcal{O}(K \times \widetilde{N}) + \mathcal{O}(K \times \widetilde{N} \times n \times (T_{clip} + T_{adaptor}))$ and the time complexity of a pure sub-dimensional dense retriever is $\mathcal{O}(N \times n \times (T_{clip} + T_{adaptor}))$.*

*Proof.* The formal proof can be found in Appendix B. □

Since $\widetilde{N} \ll N$ and $K$ is a relatively small constant, the dominant term of Algorithm 1 is far less than a pure sub-dimensional dense retriever. In terms of retrieval efficiency, we adopt Algorithm 1 - a hybrid of sub-dimensional sparse and dense retriever.

**Theorem 3.2** (Algorithm Optimality). *Let $\delta_{\min} = \min\{\alpha_i \mid \alpha_i > 0\}$ be the smallest nonzero subquery weight, and $C_{\max} = \max \sum_{i=1}^{n} \cos(v_{j,i}, t_i)$. For any $0 < \beta < \beta_{max} = \frac{\delta_{min}}{C_{\max}}$, Algorithm 1 returns all Pareto-optimal solutions to Eq.6.*

*Proof.* The formal proof can be found in Appendix C. □

### 3.2 IMAGE GENERATION WITH RETRIEVED IMAGES

To generate an image from the user query $Q$ while ensuring that the *satisfied* subquery features in $\mathcal{P}$ are preserved, we utilize a pretrained MLLM with subquery-aware instructions.

Given the set of retrieved images is $\mathcal{P} = \{I_j^*\}$, each retrieved image $I_j^*$ is associated with a subquery satisfaction vector $\mathbf{s}(I_j^*)$. Let

$$Q_j = \{q_i \mid s_i(I_j^*) = 1\} \tag{9}$$

be the subset of subqueries from the user query $Q$ that $I_j^*$ actually satisfies.

For each image $I_j^* \in \mathcal{P}$, we construct an in-context example in a form: $\langle I_j^* \rangle$ *Use only [$Q_j$] in [$I_j^*$].*

Here, $\langle I_j^* \rangle$ denotes the visual tokens for the $j$-th retrieved image, [$Q_j$] is the satisfied subqueries in $I_j^*$, and [$I_j^*$] is "the $j$-th retrieved image".

Next, we feed the in-context examples to a pretrained MLLM together with the original query $Q$. The MLLM, which operates in an autoregressive manner, is thus guided to generate the final image $\hat{I}$ as:

$$p_\theta\big(\hat{I} \mid Q, \{I_j^*\}, \{Q_j\}\big) = \prod_{t=1}^{T} p_\theta\Big(\hat{I}_t \mid \hat{I}_{<t}, Q, \{I_j^*\}, \{Q_j\}\Big), \tag{10}$$

where $\hat{I}_t$ denotes the $t$-th visual token in the generated image representation, and $\theta$ represents the parameters of the pretrained MLLM. By referencing the full prompt: `[Q]` $\langle I_j^* \rangle$ `Use only [`$Q_j$`] in [`$I_j^*$`]`, the MLLM learns to preserve the relevant subquery features that each retrieved image $I_j^*$ contributes.

## 4 EXPERIMENTS

### 4.1 EXPERIMENT SETUP

**Baselines and Evaluation Metrics** We compare our proposed method with several baselines on text-to-image retrieval and text-to-image generation.

- Text-to-Image Retrieval Baselines: CLIP (ViT-L/14) (Radford et al., 2021) is a dual-encoder model pretrained on large-scale image-text pairs and remains the most widely adopted baseline for T2I retrieval. SigLIP (ViT-SO400M/14@384) (Zhai et al., 2023) improves retrieval precision over CLIP by replacing the contrastive loss with a sigmoid-based loss. ViLLA (Varma et al., 2023) is a large-scale vision-language pretraining model with multi-granularity objectives. BLIP-2 is an efficient vision-language model (Li et al., 2023). We report BLIP-2 results as cited from (Ge et al., 2024). GILL (Koh et al., 2023) is a unified framework that combines retrieval and generation.

- Text-to-Image Generation Baselines: SDXL (Podell et al., 2023) is a widely used high-quality T2I diffusion model. FLUX.1-dev (Labs, 2024) is a T2I DiT-based model with high-fidelity outputs. LaVIT (Jin et al., 2023) is a vision-language model that supports T2I generation. RDM (Blattmann et al., 2022) is a representative retrieval-augmented diffusion model. UniRAG (Sharifymoghaddam et al., 2024) is a recent retrieval-augmented vision-language model. GILL (Koh et al., 2023) can perform both T2I-R and T2I-G. We use gpt-image-1 (OpenAI, 2025) and gemini-2.0-flash (Google, 2025) as our MLLM backbones, and compare them against their MLLM baselines without retrieval.

Table 1: Evaluation of Text-to-Image Retrieval on MS-COCO and Flickr30K.

| Method | MS-COCO (5K) | | | Flickr30K (1K) | | |
|---|---|---|---|---|---|---|
| | R@1 | R@5 | R@10 | R@1 | R@5 | R@10 |
| CLIP (ViT-L/14) | 43.26 | 68.70 | 78.12 | 77.40 | 94.80 | 96.60 |
| Finetuned CLIP (ViT-L/14) | 46.10 | 72.52 | 82.20 | 78.50 | 95.00 | 97.30 |
| SigLIP | 46.96 | 71.72 | 80.64 | 82.20 | 95.90 | 97.70 |
| ViLLA | 34.77 | 60.67 | 70.69 | 59.41 | 85.02 | 92.82 |
| BLIP-2 | 59.10 | 82.70 | 89.40 | 82.40 | 96.50 | 98.40 |
| GILL | 32.12 | 57.73 | 66.55 | 55.41 | 81.94 | 89.77 |
| **Ours** | **81.82** | **97.46** | **99.38** | **97.50** | **100.00** | **100.00** |

For T2I-R, we adopt the standard retrieval metric Recall at K(R@K, K=1, 5, and 10). For T2I-G, we evaluate the quality of generated images by computing the average pairwise cosine similarity of generated and ground-truth images with CLIP(ViT-L/14) (Radford et al., 2021), DINOv2(ViT-L/14) (Stein et al., 2023), and SigLIP(ViT-SO400M/14@384) (Zhai et al., 2023) embeddings. We also employ style loss (Gatys et al., 2015) to assess artistic style transfer in WikiArt (Ushio, 2024).

**Dataset Construction** We evaluate the text-to-image retrieval on the standard benchmark MS-COCO (Chen et al., 2015) and Flickr30K (Young et al., 2014) test sets. As for the text-to-image generation, we evaluate the model's image generation capabilities across different aspects and choose three datasets: artistic style transfer in the WikiArt (Ushio, 2024), fine-grained image generation in the CUB (Wah et al., 2011), and long-tailed image generation in the ImageNet-LT (Liu et al., 2019). For each generation dataset, we select some test samples and use the remaining as the retrieval database. More details in Appendix D.

**Implementation Details** For T2I-R, our sub-dimensional dense retriever is composed of a pretrained CLIP vision encoder (ViT-L/14) and an adaptor. We train the sub-dimensional dense retriever on the COCO training set using the InfoNCE loss with a temperature of 0.07. The adaptor is optimized using the Adam optimizer with an initial learning rate of 5e-5, and a StepLR scheduler with a step size of 3 epochs and a decay factor of 0.6. The finetuned CLIP (ViT-L/14) is finetuned on the same COCO training set. We set $\beta = 0.015$ based on Therorem 3.2. The experiments[1] are conducted on a 64-bit machine with 24-core Intel 13th Gen Core i9-13900K@5.80GHz, 32GB memory and NVIDIA GeForce RTX 4090.

### 4.2 QUANTITATIVE EVALUATION OF TEXT-TO-IMAGE RETRIEVAL

We test our sub-dimensional dense retriever model with various types of T2I-R models. As shown in Tab. 1, our proposed sub-dimensional dense retriever achieves state-of-the-art performance across all metrics and outperforms all baselines by a substantial margin on both MS-COCO and Flickr30K datasets. On MS-COCO, our method achieves R@1 = 81.82%, R@5 = 97.46%, and R@10 = 99.38%, which are significantly higher than the best-performing baseline BLIP-2. The relative improvements are 38.4% on R@1, 17.8% on R@5, and 11.2% on R@10, demonstrating our model's superior capability in text-to-image retrieval. Notably, it exhibits strong zero-shot T2I-R performance on Flickr30K, achieving near-perfect accuracy with R@1 = 97.50%, R@5 = 100.00%, and R@10 = 100.00%, and surpassing CLIP's R@1 = 77.40% by nearly 26%. These results confirm that our proposed sub-dimensional dense retriever significantly enhances fine-grained T2I-R compared to global embedding alignment such as CLIP, finetuned CLIP, SigLIP, and BLIP-2, region-based fine-grained match on ViLLA, and the retrieval and generation unified framework GILL.

### 4.3 QUANTITATIVE EVALUATION OF TEXT-TO-IMAGE GENERATION

We benchmark our Cross-modal RAG method against state-of-the-art text-to-image generation models, including diffusion-based (SDXL, FLUX), autoregressive (LaVIT, gemini-2.0-flash, gpt-image-1), retrieval-augmented (RDM, UniRAG), and retrieval and generation unified (GILL) baselines. The evaluation is conducted on three datasets that span different generation challenges: WikiArt (artistic style transfer), CUB (fine-grained), and ImageNet-LT (long-tailed). As shown in Tab. 2, our Cross-modal RAG achieves the highest scores across all models and datasets, outperforming both its MLLM backbone without retrieval and other text-to-image generation models. These results demonstrate that our retrieval module is highly generalizable across different MLLM backbones and more effective than existing retrieval-augmented methods. On WikiArt, our method achieves the best performance

---

[1]The code is available at https://anonymous.4open.science/r/Cross-modal-RAG-3DE8.

Table 2: Evaluation of Text-to-Image Generation on WikiArt, CUB, and ImageNet-LT.

| Method | WikiArt | | | | CUB | | | ImageNet-LT | | |
|---|---|---|---|---|---|---|---|---|---|---|
| | CLIP ↑ | DINO ↑ | SigLIP ↑ | Style Loss ↓ | CLIP ↑ | DINO ↑ | SigLIP ↑ | CLIP ↑ | DINO ↑ | SigLIP ↑ |
| FLUX.1-dev | 0.605 | 0.307 | 0.617 | 0.056 | 0.720 | 0.372 | 0.681 | 0.654 | 0.381 | 0.625 |
| SDXL | 0.688 | 0.504 | 0.720 | 0.022 | 0.743 | 0.519 | 0.738 | 0.668 | 0.403 | 0.653 |
| LaVIT | 0.689 | 0.485 | 0.721 | 0.036 | 0.676 | 0.245 | 0.647 | 0.662 | 0.365 | 0.652 |
| RDM | 0.507 | 0.237 | 0.528 | 0.024 | 0.638 | 0.326 | 0.663 | 0.576 | 0.333 | 0.603 |
| UniRAG | 0.646 | 0.362 | 0.654 | 0.068 | 0.746 | 0.344 | 0.718 | 0.610 | 0.255 | 0.600 |
| GILL | 0.629 | 0.439 | 0.654 | 0.027 | 0.719 | 0.185 | 0.675 | 0.635 | 0.228 | 0.615 |
| gemini-2.0-flash | 0.721 | 0.537 | 0.723 | 0.026 | 0.726 | 0.569 | 0.713 | 0.692 | 0.456 | 0.682 |
| gpt-image-1 | 0.730 | 0.542 | 0.733 | 0.024 | 0.735 | 0.575 | 0.708 | 0.695 | 0.476 | 0.683 |
| **Ours (gemini-2.0-flash)** | 0.735 | 0.568 | 0.738 | 0.021 | 0.760 | 0.595 | **0.747** | **0.825** | 0.717 | **0.814** |
| **Ours (gpt-image-1)** | **0.746** | **0.604** | **0.744** | **0.019** | **0.764** | **0.600** | 0.744 | 0.815 | **0.761** | 0.812 |

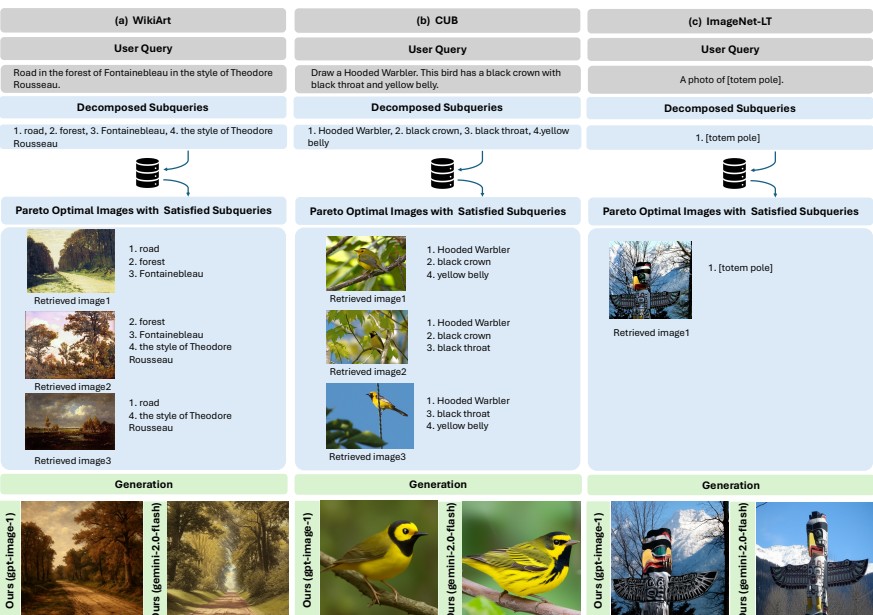

Figure 3: The retrieved pareto optimal images with their corresponding satisfied subqueries in (a) WikiArt, (b) CUB and (c) ImageNet-LT datasets and generation results of Cross-modal RAG.

in CLIP, DINO, and SigLIP, along with the lowest style loss, indicating it can capture the particular artistic style specified in the retrieved images effectively. On CUB, Cross-modal RAG also performs strongly across all three metrics, because it can localize and leverage the specific visual details in the retrieved images to facilitate generation. On ImageNet-LT, our method can retrieve images that best match the query, which greatly benefits T2I-G in the long-tailed situation.

## 4.4 QUALITATIVE ANALYSIS

To qualitatively illustrate the effectiveness of our Cross-modal RAG model, we visualize some examples of our retrieved pareto optimal images with their corresponding satisfied subqueries and generated outputs across all datasets in Fig. 3. The satisfied subqueries of each retrieved Pareto-optimal image are non-overlapping, and each retrieved image is optimal with respect to the sub-dimensions it satisfies. Therefore, we can guarantee that the Pareto set $\mathcal{P}$ collectively covers images with all satisfied subqueries in the retrieval database $\mathcal{D}$. Moreover, since the model knows which subqueries are satisfied by each retrieved image, MLLMs can be guided to condition on the relevant subquery-aligned parts of each retrieved image during generation. As shown Fig. 3(a), the model is capable of *style transfer*, learning the artistic style of a certain artist (*e.g.*, Theodore Rousseau) while preserving the details corresponding to each subquery (*e.g.*, road, forest, Fontainebleau). The model is also able to retrieve accurate fine-grained information and compose the entities in subqueries (*e.g.*, black crown, black throat, yellow belly) to perform *fine-grained image generation* on the CUB dataset in Fig. 3(b). Moreover, the model is good at *long-tailed or knowledge-intensive image generation*. In Fig. 3(c), ImageNet-LT is a long-tailed distribution dataset with many rare entities (*e.g.*, totem pole). Retrieving such correct images can help improve generation fidelity. Baseline models without

Table 3: Evaluation of the retrieval efficiency on the COCO. Our methods are denoted in gray .

| Method | GPU Memory (MB) | # of Parameters (M) | Query Latency (ms) |
|---|---|---|---|
| CLIP (ViT-L/14) | 2172.19 | 427.62 | 8.68 |
| dense (all) | 2195.26 | 433.55 | 14.22 |
| dense (adaptor) | 23.07 | 7.31 | 4.35 |
| sparse | 0 | 0 | 2.17 |

retrieval capabilities tend to struggle in these scenarios. More comparisons of generated images with other baselines are provided in Appendix H.

### 4.5 EFFICIENCY ANALYSIS

As shown in Tab. 3, we compare the retrieval efficiency of CLIP (ViT-L/14) with our sub-dimensional dense and sparse retrievers on the COCO test set. Our sub-dimensional dense retriever is composed of a frozen CLIP encoder (ViT-L/14) and a lightweight adaptor. As reported in Table 1, the sub-dimensional dense retriever improves Recall@1 by +89.14% over CLIP on COCO. Despite the adaptor's minimal overhead – only 0.01× CLIP's GPU memory usage, 0.017× its number of parameters, and 0.5× its query latency – its performance gain is substantial. Our sub-dimensional sparse retriever is text-based and operates solely on the CPU, requiring no GPU memory consumption, no learnable parameters and achieving the lowest query latency. Our Cross-modal RAG method, a hybrid of our sub-dimensional sparse and dense retriever, can leverage the complementary strengths of both and achieve query latency that lies between the pure sparse and dense retrievers – closer to that of the sparse. These results show Cross-modal RAG's efficiency and scalability for large-scale text-to-image retrieval tasks without compromising effectiveness.

### 4.6 ABLATION STUDY

**Ablation Study on Subquery Decomposition**

We evaluate retrieval performance without subquery decomposition on multi-subquery datasets, WikiArt and CUB, by directly using a BM25 retriever to retrieve the top-1, top-2, and top-3 images based on the full user query. Our multi-objective joint retrieval method achieves a higher subquery coverage rate compared to the conventional text-based BM25 retrieval on both WikiArt and CUB in Fig. 4. This result indicates that our multi-objective joint retrieval

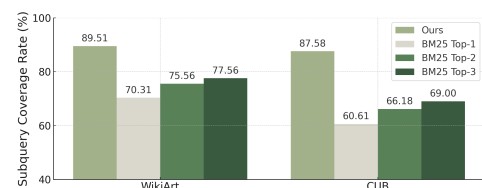

Figure 4: Ablation Study on Subquery Decomposition on the WikiArt and CUB.

method retrieves a set of images $\mathcal{P}$ that collectively cover the largest number of subqueries from $\mathcal{D}$, demonstrating its superior ability to capture the full semantic intent of the user query.

**Ablation Study on the Sub-dimensional Dense Retriever**

We retain the sub-dimensional sparse retriever and replace the sub-dimensional dense retriever in Cross-modal RAG with a randomly selected image. The results in Tab. 4 show that our dense retriever is able to retrieve images that best match the entity in the query in the ImageNet-LT. Notably, our dense retriever, though trained only on the COCO, generalizes well to unseen entities on the ImageNet-LT.

Table 4: Ablation Study of our Cross-modal RAG w/o dense retriever on the ImageNet-LT.

| Method | CLIP ↑ | DINO ↑ | SigLIP ↑ |
|---|---|---|---|
| Ours (gpt-image-1) | 0.815 | 0.761 | 0.812 |
| w/o dense | 0.773 | 0.607 | 0.752 |

### 5 CONCLUSION

We proposed Cross-modal RAG, a novel sub-dimensional text-to-image retrieval-augmented generation framework. By efficient and fine-grained T2I retrieval, it facilitates domain-specific, fine-grained, and long-tailed image generation. Our method leverages a hybrid retrieval strategy combining sub-dimensional sparse filtering with dense retrieval to precisely align subqueries with visual elements, guiding a MLLM to generate coherent images on the subquery level. The Pareto-optimal image selection ensures the largest coverage of various aspects in the query. Extensive experiments demonstrated Cross-modal RAG's superior performance over state-of-the-art baselines in T2I-R and can also benefit T2I-G. The ablation study and efficiency analysis highlight the effectiveness of each component and efficiency in the Cross-modal RAG.

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

## A  DECOMPOSING QUERIES $Q$ INTO SUBQUERIES $q_i$

We decompose queries $Q$ into subqueries $q_i$ using the prompt template in Fig. 5 to obtain $t_i$. Furthermore, we remove the attribute modifiers in $q_i$ to derive $T_i$ with an LLM (e.g., GPT-4o mini). For example, if $t_i$ corresponds to the embedding of "orange cat", then $T_i$ corresponds to the embedding of "cat". In some cases where $q_i$ does not contain any attribute modifiers, $t_i$ and $T_i$ are identical.

---

**Decomposing Queries into Subqueries**

Given an image caption, decompose the caption into an atomic entity. Each entity should preserve descriptive details (e.g., size, color, material, location) together with the entity in a natural, readable phrase. The entity should contain a noun and reserve noun modifiers in the caption. Please ignore the entities like 'a photo of', 'an image of', 'an overhead shot', 'the window showing' that are invisible in the image and ignore the entities like 'one' and 'the other' that have duplicate entities before.

Caption: two cars are traveling on the road and waiting at the traffic light.
Entity: cars, road, traffic light

Caption: duplicate images of a girl with a blue tank top and black tennis skirt holding a tennis racquet and swinging at a ball.
Entity: girl, blue tank top, black tennis skirt, tennis racqet, ball

Caption: the window showing a traffic signal is covered in droplets of rainwater.
Entity: traffic signal, droplets of rainwater

Caption: an overhead shot captures an intersection with a "go colts" sign.
Entity: intersection, "go colts" sign

Caption: a van with a face painted on its hood driving through street in china.
Entity: van, a face painted on its hood, street in china

Caption: two men, one with a black shirt and the other with a white shirt, are kicking each other without making contact.
Entity: men, black shirt, white shirt

Caption: {*caption*}
Entity:

---

Figure 5: The Prompt Example for Decomposing User Queries into Subqueries on MS-COCO.

## B  PROOF OF THE TIME COMPLEXITY IN RETRIEVAL EFFICIENCY

1. Proof of the time complexity for Algorithm 1

   Let $N$ be the total number of images in $D$. We can score each image's sparse textual match in $\mathcal{O}(N)$. We discard images that do not satisfy any subquery, leaving a reduced set $\widetilde{D} \subseteq D$ of size $\widetilde{N}$.

   We then discretize the simplex of subquery weights $\alpha$ into $K$ possible combinations. Each combination requires checking $\sum_i \alpha_i s_i(I_j)$ in $\mathcal{O}(\widetilde{N})$ time , thus $\mathcal{O}(K \times \widetilde{N})$ in total.

   Each adaptor pass handles both CLIP-based vision encoding (costing $T_{\text{clip}}$ and the adaptor's own cross-attention (costing $T_{\text{adaptor}}$)). If we assume one pass per subquery set (of size $n$), the total cost is $\widetilde{N} \times n \times (T_{\text{clip}} + T_{\text{adaptor}})$. Multiplied by $K$ weight vectors, this yields $\mathcal{O}\left(K \times \widetilde{N} \times n \times (T_{\text{clip}} + T_{\text{adaptor}})\right)$.

   Combining the steps above, total time is:
   $$\mathcal{O}(N) + \mathcal{O}(K \times \widetilde{N}) + \mathcal{O}\left(K \times \widetilde{N} \times n \times (T_{\text{clip}} + T_{\text{adaptor}})\right).$$

2. Proof of the time complexity for a pure sub-dimensional dense retriever

If we skip the sparse filter, we must embed all $N$ images for each subquery. Thus, the pure dense approach demands $\mathcal{O}\left(N \times n \times (T_{\text{clip}} + T_{\text{adaptor}})\right)$.

Because $\widetilde{N} \ll N$ and $K$ is small, this total is typically far lower than scanning all $N$ images with the sub-dimensional dense retriever.

## C  PROOF OF ALGORITHM OPTIMALITY

Because $s_i(I_j) \in \{0, 1\}$ and $\sum_i \alpha_i = 1$, $\sum_i \alpha_i s_i(I_j)$ lies in $[0, 1]$. Since $\cos(v_{j,i}, t_i) \in [0, 1]$, $\sum_i \cos(v_{j,i}, t_i) \leq n$. Hence $C_{\max} \leq n$.

Suppose $I_a$ dominates $I_b$. Then

$$\Delta_{\text{sparse}} = \sum_i \alpha_i s_i(I_a) - \sum_i \alpha_i s_i(I_b) > 0. \tag{11}$$

Let

$$\Delta_{\text{dense}} = \sum_i \cos(v_{a,i}, t_i) - \sum_i \cos(v_{b,i}, t_i). \tag{12}$$

We want $\Delta_{\text{sparse}} + \beta \Delta_{\text{dense}} > 0$. In the worst case for $I_a$, $\Delta_{\text{dense}} < 0$, potentially as low as $-C_{\max}$. A sufficient condition for $I_a$ to stay preferred is

$$\Delta_{\text{sparse}} - \beta C_{\max} > 0.$$

Because $\Delta_{\text{sparse}} \geq \delta_{\min}$ if $I_a$ indeed satisfies at least one more subquery and $\beta > 0$ is assumed by definition, we obtain:

$$0 < \beta < \frac{\delta_{\min}}{C_{\max}} = \beta_{max}. \tag{13}$$

We discretize the simplex $\{\alpha : \alpha_i \geq 0, \sum_i \alpha_i = 1\}$. Because subqueries are strictly enumerated by $\alpha$, if an image satisfies a unique subquery set, it must appear as an $\arg\max \sum_i \alpha_i s_i(I_j)$ for some $\alpha$. Thus, no non-dominated $\mathbf{s}(I_j)$ is missed. $\Delta_{\text{dense}}$ can be further used to find an optimal image among those sharing the same $\mathbf{s}(I_j)$. Therefore, all Pareto-optimal solutions are obtained and $\mathbf{s}(I_j)$ in the Pareto front $\mathcal{P}_f$ is unique.

## D  DATASETS

For T2I-R on MS-COCO, we follow the Karpathy split using 82,783 training images, 5,000 validation images, and 5,000 test images. For Flickr30K, we only use 1,000 images in the test set for evaluation.

Regarding T2I-G, WikiArt dataset is a comprehensive collection of fine art images sourced from the WikiArt online encyclopedia. Our implementation is based on the version provided in (Ushio, 2024). To construct the test set, we compare artwork titles across different images and identify pairs that differ by at most three tokens. From each matched pair, we retain one sample, resulting in 2,619 distinct test examples. The query for each test sample is formatted as: *<title> in the style of <artistName>*. The retrieval database is composed of the remaining WikiArt images after excluding all test samples, ensuring no overlap between ground-truth and retrieval candidates, as shown in Tab. 5. The Caltech-UCSD Birds-200-2011 (CUB-200-2011) (Wah et al., 2011) is a widely used benchmark for fine-grained image classification and generation tasks. It contains 11,788 images across 200 bird species. We use the CUB dataset with 10 single-sentence visual descriptions per image collected by (Reed et al., 2016). Similarly, to construct the test set, we compare captions across different images and identify pairs that differ by one token, resulting in 5,485 distinct test samples. The query for each test sample is formatted as: *Draw a <speciesName>. <caption>*. For each test sample, the retrieval candidates consist of all remaining images in the CUB dataset, excluding that test image. The ImageNet-LT dataset (Liu et al., 2019) is a long-tailed version of the original ImageNet dataset. It contains 1,000 classes with 5 images per class. We randomly choose one image from each class to construct the test samples. The retrieval database is composed of the remaining ImageNet-LT images after excluding all test samples. The query for each test sample is formatted as: *A photo of <className>*.

Table 5: Data construction of the T2I-G datasets

| Dataset | # of image in the dataset | # of test samples | # of images in the retrieval database |
|---------|---------------------------|-------------------|----------------------------------------|
| WikiArt | 63,061 | 2,619 | 60,442 |
| CUB | 11,788 | 5,485 | 11,787 |
| ImageNet-LT | 50,000 | 1,000 | 49,000 |

## E    LIMITATIONS

While our multi-objective joint retrieval combining sub-dimensional sparse and dense retrievers is efficient and achieves good granularity, if the image retrieval database $\mathcal{D}$ does not contain any images relevant to the query, the retrieval cannot provide benefits to the generation. However, this drawback is not an issue unique to our RAG model; rather, the success of any RAG approach depends on the presence of relevant data (i.e., images in our case) in the database. As long as there exist partially related images in the database—even if they do not perfectly match all aspects of the query—our method can identify the most overlapping images with the query to effectively support generation.

## F    LLM USAGE DISCLOSURE

We use large language models to correct the grammar and improve the clarity of writing in this paper.

## G    HYPER-PARAMETER ANALYSIS

For the trade-off parameter $\beta$ and the weight vector $\alpha$, in our experiments, we set $\beta = 0.015$ based on Theorem 3.2, where for any $0 < \beta < \beta_{\max} = \frac{\delta_{\min}}{C_{\max}}$, our algorithm guarantees to return all Pareto-optimal solutions. The maximum number of subqueries is 8 in our experiments. Assuming equal weight for each subquery, we have $\beta_{\max} = \frac{1/8}{8} \approx 0.0156$, so we set $\beta = 0.015$ accordingly.

The design of $\alpha$ is intended to ensure the selection of a Pareto-optimal set of images would collectively cover all subqueries in the input user query. If an image satisfies a subquery that is not covered by other images, then there exists at least one choice of $\{\alpha_i\}$ for which it can maximize $\sum_{i=1}^{n} \alpha_i s_i(I_j)$, allowing that image to be selected. Therefore, in our experiments, we vary $\alpha$ across all possible extreme values to ensure that every possible combination of subqueries is taken into account.

## H    MORE VISUALIZATIONS OF OUR METHOD COMPARED WITH OTHER BASELINES

More visualizations of our proposed method Cross-modal RAG compared with other baselines can be found in Fig. 6 to 8. Across all three datasets, our method achieves superior image generation capability. For the CUB in Fig. 6, our Cross-modal RAG can generate realistic images that align with all subqueries, which are often ignored or distorted in GILL and UniRAG. Besides, SDXL, LaVIT, and RDM tend to generate the sketch-like images rather than photo-realistic birds. On the ImageNet-LT in Fig. 7, retrieving relevant images plays a crucial role in generating accurate long-tailed objects. Our method successfully generates all three long-tailed objects with high visual fidelity. In contrast, none of the baselines are able to generate all three correctly - UniRAG and GILL even fail to produce a single accurate image. For WikiArt in Fig. 8, creative image generation poses a unique challenge, as it is inherently difficult to reproduce the exact ground-truth image. However, our method explicitly retrieve the images with satisfied subqueries and can capture the particular artistic style specified in the query. As a result, all six generated images of Cross-modal RAG closely resemble the style of the target artist in the query. In contrast, other RAG baselines can not guarantee if the retrieved images grounded in the intended artist's style. RDM even suffers from low visual fidelity when generating human faces.

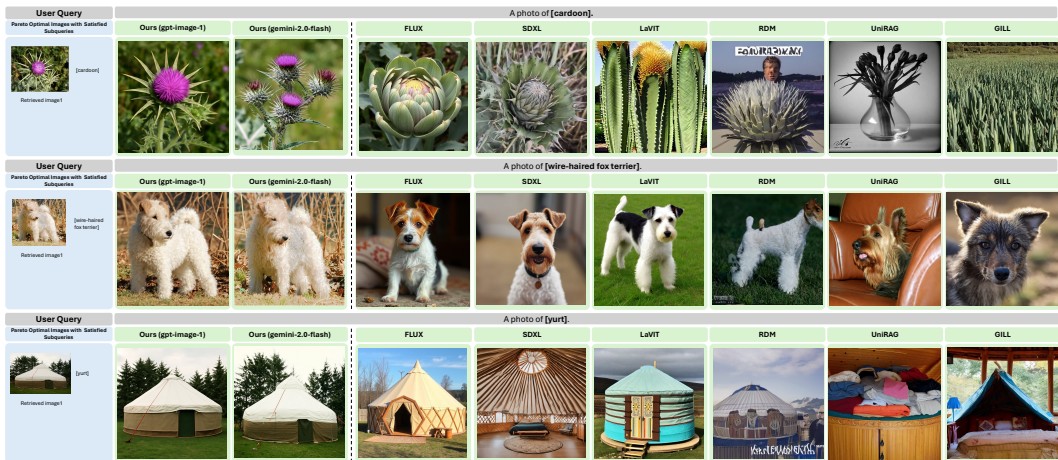

Figure 6: Visualizations of generation on CUB compared with other baselines.

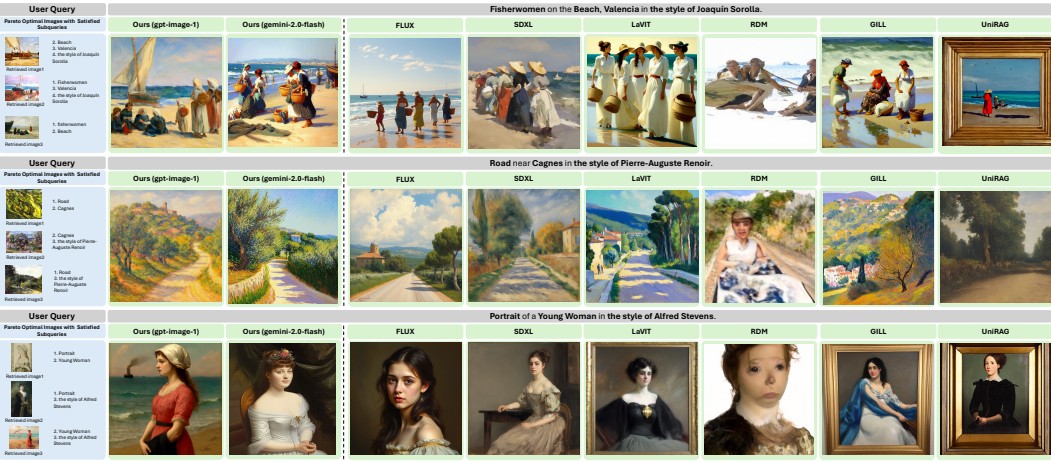

Figure 7: Visualizations of generation on ImageNet-LT compared with other baselines.

Figure 8: Visualizations of generation on WikiArt compared with other baselines.

