# OpenReview forum: "Cross-modal RAG: Sub-dimensional Text-to-Image Retrieval-Augmented Generation"
_ICLR.cc/2026/Conference — Submitted to ICLR 2026_

### Official Review · Reviewer_jqS4 · 2025-10-30

**Soundness:** 3
**Presentation:** 3
**Contribution:** 3
**Rating:** 4
**Confidence:** 4

**Summary:**

Cross-modal RAG addresses a core limitation of existing text-to-image RAG systems: global image-text similarity often fails when complex queries span multiple fine-grained aspects and no single image satisfies all constraints. The paper proposes a sub-dimensional framework that decomposes both the textual query into subqueries and candidate images into aligned visual sub-dimensions. Retrieval is cast as a multi-objective problem that jointly balances lexical subquery satisfaction (sparse stage) with fine-grained semantic alignment (dense stage) to select a Pareto-optimal set of complementary images. Generation is then guided by a MLLM with subquery-aware instructions so that only the relevant visual components from each retrieved image condition the final synthesis.

**Strengths:**

The paper advances RAG for text-to-image generation by dual decomposition—subqueries on the textual side and sub-dimensional visual embeddings on the image side—which enables targeted alignment at a finer granularity than global CLIP-like matching. The multi-objective formulation and Pareto-optimal image selection are conceptually elegant and differ from common top-k retrieval pipelines. Compared to fine-grained retrieval works that crop regions or use patch-level alignment, the proposed adaptor derives sub-dimensional embeddings without explicit region cropping and aligns them to subqueries, which is both efficient and robust to noisy region proposals. The subquery-aware generation via MLLM prompting (“Use only [Qj] in [Ij]”) is a pragmatic and model-agnostic way to control conditioning at inference.

**Weaknesses:**

1. Reliance on dataset captions for the sparse stage raises questions about robustness in settings where captions are incomplete, noisy, or absent. While the dense stage helps, the algorithm filters to by sparse satisfaction, which could discard relevant images that use synonyms or paraphrases not captured by lexical matching. The approach assumes successful query decomposition—errors or ambiguities in subquery extraction (or in converting ti to core concepts Ti) can propagate to both retrieval and generation.

2. Lack of ablation study on evaluation of text-to-image generation on WikiArt, CUB, and ImageNet-LT. The authors are advised to provide more results under the settings of w/o RAG, w/o Stage 1, w/o Stage 2, w/o Stage 3, etc.

**Questions:**

1. What is the sensitivity of retrieval and generation quality to β and the α schedule? Beyond the theoretical bound, does a learned α improve coverage/quality for user-specific priorities?

2. How reliable is subquery decomposition across domains? Do you have error analyses showing typical failure modes (entity conflation, attribute omission), and how the dense stage compensates when decomposition is imperfect?

3. How large can the Pareto set P be in practice, and how does generation quality/latency scale with the number of in-context images? Are there diminishing returns beyond 2–3 images, and do you prune overlapping subquery coverage proactively?

---

> ### Author Response · Authors · 2025-11-22
>
> We sincerely appreciate that the reviewer found our multi-objective formulation and Pareto-optimal image selection are conceptually elegant and differ from common top-k retrieval pipelines and fine-grained retrieval works that crop regions or use patch-level alignment. Please refer to our response below for details:
>
> > W1: Reliance on dataset captions for the sparse stage raises questions about robustness in settings where captions are incomplete, noisy, or absent.
>
> A1: We appreciate the reviewer's comment. In practice, we could further enhance caption robustness by applying query expansion (caption expansion) to correct noisy captions and to expand captions by incorporating missing visual content. Even with the noisy captions, our method already achieves state-of-the art text-to-image retrieval performance on MS-COCO and Flickr30K, indicating that the approach is not brittle in realistic caption quality regimes. We currently do not incorporate query expansion to ensure fairness with other baselines.
>
> > W2: Lack of ablation study on evaluation of text-to-image generation on WikiArt, CUB, and ImageNet-LT.
>
> A2: We thank the reviewer for the suggestion. In Table2, the row “gpt-image-1” already serves as the w/o RAG baseline. In addition, we have now included ablation results using gpt-image-1 as the backbone under w/o Stage 1 (sub-dimensional sparse retriever) and w/o Stage 2 (sub-dimensional dense retriever). As for Stage 3, it is not an independent module but the multi-objective joint retrieval formulation that combines sparse and dense; removing it degenerates exactly to using either only the sparse or only the dense retriever, which is already covered as follows.
>
> | Method        |        |        |    WikiArt        |               |           |     CUB    |           |          |     ImageNet-LT    |           |
> |---------------|:---------------:|:--------:|:-----------:|:----------------:|:---------------:|:--------:|:-----------|----------------------:|:--------:|:-------:|
> |                      | CLIP↑        | DINO↑| SigLIP↑ | Style Loss↓&nbsp;&nbsp; | CLIP↑       | DINO↑ | SigLIP↑&nbsp;&nbsp;&nbsp;&nbsp; | CLIP↑ | DINO↑ | SigLIP↑ |
> **Ours (gpt-image-1)** | **0.746** | **0.604** | **0.744** | **0.019** | **0.764** | **0.600** | **0.744** | **0.815** | **0.761** | **0.812** |
> &nbsp;&nbsp; w/o RAG | 0.730 | 0.542 | 0.733 | 0.024 | 0.735 | 0.575 | 0.708 | 0.695 | 0.476 | 0.683 |
> &nbsp;&nbsp; w/o stage 1 |	0.735 | 0.551 | 0.735 | 0.023 | 0.740 | 0.580 | 0.717 | 0.720 | 0.502 | 0.718 |
> &nbsp;&nbsp; w/o stage 2 | 0.740 | 0.598 | 0.741 | 0.021 | 0.757 | 0.596 | 0.740 | 0.773 | 0.607 | 0.752|
>
> > Q1: What is the sensitivity of retrieval and generation quality to $\beta$ and the $\alpha$ schedule? Beyond the theoretical bound, does a learned $\alpha$ improve coverage/quality for user-specific priorities?
>
> A3: The parameters of $\alpha$ and $\beta$ only affect the retrieval objective in Eq.6, and influence generation indirectly through which images are retrieved. As Theorem 3.2 specifies, $\alpha$ enumerates all possible extreme values, and for any $0<\beta<\beta_{max}=\frac{\delta_{min}}{C_{\max}}$, the method is guaranteed to return all Pareto-optimal images. Therefore, when $\beta$ lies within this valid range, our method is robust with respect to both retrieval and generation quality.
>
> The design of $\alpha$ is intended to ensure the selection of a Pareto-optimal set of images would collectively cover all subqueries in the input user query. If an image satisfies a subquery that is not covered by other images, then there exists at least one choice of \{$\alpha_i$\} for which it can maximize $ \sum_{i=1}^{n} \alpha_i s_i(I_j)$, allowing that image to be selected. If the user has specific priorities, we can also adjust $\alpha$ to ensure images satisfying specific subqueries are selected and ranked top.
>
> > Q2: How reliable is subquery decomposition across domains?
>
> A4: Our subqueries are extracted by decomposing the user query with an off-the-shelf LLM (e.g., GPT-4o mini) using a structured prompt. To assess the reliability of this process, we compute the parsing accuracy, defined as the percentage of subqueries that can be directly matched to substrings in the original query. The results show that the subquery parsing accuracy is 99.70\% on WikiArt, 98.64\% on CUB, and 100\% on ImageNet-LT. We further manually examined all failed cases, and found that in every case, the subquery actually corrected a typo in the original user query (i.e., a typo in the original caption provided by the dataset). For example, one original caption in the CUB dataset is "This bird is white and brown n color, with a black beak." and the extracted subquery correctly fixes the typo as "This bird is white and brown in color, with a black beak.". Given the extremely low error rate and the fact that the mismatches actually enhance the quality of the input, we found that our subquery decomposition is reliable in practice.

---

> ### Author Response · Authors · 2025-11-22
> **Official Comment by Authors (cont'd)**
>
> > Q3: How large can the Pareto set P be in practice, and how does generation quality/latency scale with the number of in-context images? Are there diminishing returns beyond 2–3 images, and do you prune overlapping subquery coverage proactively?
>
> A5: The Pareto set P is often 3-5 images in practice. Regarding the concern about overlapping subquery coverage, our algorithm is designed to inherently prune such redundancy. As described in the qualitative analysis (Section 4.4), "the satisfied subqueries of each retrieved Pareto-optimal image are non-overlapping, and each retrieved image is optimal with respect to the sub-dimensions it satisfies". Since the satisfied subqueries are non-overlapping, each image in the Pareto set provides unique, complementary compositional information that other images lacked. Theoretically, this further contributes to generation quality. Current MLLMs process vision tokens efficiently, and the cost of processing additional in-context images in the Pareto set is minimal compared to the generation process itself.

---

> > ### Comment · Reviewer_jqS4 · 2025-11-22
> >
> > Thank you for your response. The rebuttal addresses most of my concerns. I have raised my score accordingly.

---

> > > ### Author Response · Authors · 2025-11-22
> > >
> > > Thank you for your valuable feedback and for acknowledging our work! We appreciate your support and are glad that our responses helped address your concerns.

---

### Official Review · Reviewer_ooGi · 2025-10-30

**Soundness:** 3
**Presentation:** 3
**Contribution:** 3
**Rating:** 6
**Confidence:** 3

**Summary:**

This paper proposes Cross-modal RAG, a framework for text-to-image generation that uses sub-dimensional decomposition of queries and a hybrid retrieval strategy to handle complex prompts. It aims to improve generation by retrieving a set of complementary images and guiding an MLLM with subquery-level instructions.

**Strengths:**

1.The dual decomposition of queries and images into sub-dimensions is creative and distinguishes it from prior work.

2. ​The multi-stage framework is well-explained, and the algorithm is detailed with complexity analysis.

3. The proofs are logically structured and align well with the framework’s design.

**Weaknesses:**

1.While the authors mention FineRAG briefly in Related Work, no experimental results are provided to directly contrast the two methods.

2.The approach relies heavily on LLMs or query decomposition. Evaluating robustness to different decomposers would strengthen the work.

3. Minor Errors:

	(1) Line 249, Redundant use of “by” in the sentence: “dominated by by any other image...”

	(2) Line 277, The term “cos” appears abruptly without prior definition; it should likely align with the earlier notation "sim" (cosine similarity) introduced in Section 3 for consistency.

	(3) vji is occasionally written as vj,i (Theorem 3.2), which may cause confusion.

**Questions:**

1. See weaknesses above.

2. The selected MLLM baselines are already highly capable models. Could the observed gains be partly attributed to the inherent knowledge of these models rather than the proposed retrieval framework? To clarify this, could the authors provide ablation results where weaker generators are integrated with Cross-modal RAG?

---

> ### Author Response · Authors · 2025-11-22
>
> We sincerely appreciate that the reviewer found our paper's dual decomposition of queries and images into sub-dimensions to be creative and which distinguishes it from prior work. Below, we address the concerns raised:
>
> > W1: While the authors mention FineRAG briefly in Related Work, no experimental results are provided to directly contrast the two methods.
>
> A1: Thank you for the suggestion. However, FineRAG’s code has not been released, and the paper does not provide sufficient implementation details for reproduction. To ensure fairness and soundness, we therefore did not include FineRAG as a baseline in the experiments.
>
> > W2: The approach relies heavily on LLMs or query decomposition. Evaluating robustness to different decomposers would strengthen the work.
>
> A2: To investigate this, we implemented a rule-based decomposer using standard NLP techniques (Spacy-based Noun Phrase Chunking). While rule-based methods can identify basic noun phrases, they struggle to extract clean "atomic entities"(a noun plus its modifiers) as defined in Appendix A. They often mishandle complex attribute associations that LLMs decompose correctly. Moreover, rule-based methods frequently fail to exclude non-visual/meaningless text (e.g., retaining "a photo of" or "an image of"), which introduces noise into the retrieval process. Therefore, we find that using an LLM as the decomposer leads to more accurate decompositions that better align with our task requirements.
>
> > W3: Minor Errors.
>
> A3: We thank the reviewer for the careful proofreading. We will correct these typos in the final version as follows: (1) Removing the redundant "by" in the sentence "dominated by any other image". (2) We will replace "cos" with the previously defined "sim" (cosine similarity) for consistency. (3) We will unify the notation to $v_{ji}$ throughout the paper to avoid confusion.
>
> > Q1: Could the authors provide ablation results where weaker generators are integrated with Cross-modal RAG?
>
> A4: Thank you for your suggestion. We follow your suggestion to add a weaker generator, LaVIT. Our Cross-modal RAG framework is designed to be plug-and-play and can be applied to any MLLM-based image generator. As shown in the Table, when we integrate LaVIT with our framework, the performance consistently improves across all three datasets and all metrics. These substantial gains indicate that the observed improvements result from the effectiveness of our proposed retrieval framework.
> | Method        |        |        |    WikiArt        |               |           |     CUB    |           |          |     ImageNet-LT    |           |
> |---------------|:---------------:|:--------:|:-----------:|:----------------:|:---------------:|:--------:|:-----------|----------------------:|:--------:|:-------:|
> |                      | CLIP↑        | DINO↑| SigLIP↑ | Style Loss↓&nbsp;&nbsp; | CLIP↑       | DINO↑ | SigLIP↑&nbsp;&nbsp;&nbsp;&nbsp; | CLIP↑ | DINO↑ | SigLIP↑ |
> |LaVIT | 0.689 | 0.485 | 0.721 | 0.036 | 0.676 | 0.245 | 0.647 | 0.662 | 0.365 | 0.652 |
> |LaVIT + our retrieval | 0.723 | 0.540 | 0.730 | 0.024 | 0.724 | 0.572 | 0.720 | 0.810 | 0.668 | 0.802 |

---

### Official Review · Reviewer_fk4a · 2025-10-30

**Soundness:** 3
**Presentation:** 2
**Contribution:** 2
**Rating:** 4
**Confidence:** 4

**Summary:**

This paper proposes Cross-modal RAG, a retrieval-augmented generation framework that decomposes both queries and candidate images into sub-dimensional components for subquery-aware retrieval and image generation. The method combines a sub-dimensional sparse retriever and a dense retriever under a multi-objective optimization formulation, aiming to retrieve a Pareto-optimal set of images that collectively cover all semantic aspects of a text query. Experiments on MSCOCO, Flickr30K, WikiArt, CUB, and ImageNet-LT show notable quantitative improvements.

**Strengths:**

1. The paper presents a clear and novel perspective by modeling retrieval as a multi-objective optimization over subqueries.
2. The hybrid sparse–dense retrieval design is technically sound and efficiently implemented.
3. The paper provides extensive experimental results and clear visualizations to support the proposed idea.
4. The Pareto-optimal formulation and subquery-aware conditioning in generation are conceptually elegant and potentially generalizable.

**Weaknesses:**

1. The reported BLIP-2 results on MSCOCO deviate substantially from those in the original BLIP-2 paper, raising concerns about the fairness or correctness of baseline reproduction.
2. Evaluation on retrieval benchmarks (e.g., MSCOCO, Flickr30K) is insufficient to reflect the RAG ability for generation. These datasets primarily test retrieval performance, not retrieval-augmented reasoning or compositional synthesis, which are central to RAG.
3. The design of Stage 1 and Stage 2 is very complicated but only performs the function of query decomposition, which has been widely investigated and applied in previous works. This paper lacks comparison with naive query decomposition methods, such as leveraging an LLM as the query decomposer.

**Questions:**

same as the weaknesses.

---

> ### Author Response · Authors · 2025-11-22
>
> We thank the reviewer for acknowledging our method is technically sound and conceptually elegant. We address your questions as follows:
>
> > W1: The reported BLIP-2 results on MSCOCO.
>
> We did not re-implement BLIP-2 ourselves. Instead, we directly cite and report the BLIP-2 results in [1]. We have carefully verified that our reported BLIP-2 results on MS-COCO and Flickr30K test sets are exactly the same as the ones in Table 1 in the cited paper [1], with R@1=59.1, R@5=82.7, R@10=89.4, in text-to-image retrieval in MS-COCO and R@1=82.4, R@2=96.5, R@10=98.4 in text-to-image retrieval in Flickr30K.
>
> For BLIP-2 implementation in [1], they adopt the dual-stream image-text contrastive (ITC) for retrieval and remove the image-text matching (ITM) re-rank module in BLIP-2 for a fair comparison. Similarly, we cited BLIP-2 results from [1] because the ITM is a generic reranking module that can in principle be stacked on top of any retriever, including ours, and should therefore be disabled to compare with the retrieval module itself to ensure a fair comparison. Moreover, even if we compare against the full BLIP-2(ViT-L) results (with ITM enabled) reported in the BLIP-2 paper[2], our method still achieves significantly better performance. We will also include this clarification in our revised paper.
>
> [1] Ge, Yuying, et al. "Making LLaMA SEE and Draw with SEED Tokenizer." ICLR. 2024.
>
> [2] Li, Junnan, et al. "Blip-2: Bootstrapping language-image pre-training with frozen image encoders and large language models." International conference on machine learning. PMLR, 2023.
>
> > W2: Evaluation on retrieval benchmarks (e.g., MSCOCO, Flickr30K) is insufficient to reflect the RAG ability for generation. These datasets primarily test retrieval performance, not retrieval-augmented reasoning or compositional synthesis, which are central to RAG.
>
> A2: We would like to clarify that typical T2I RAG does not aim to perform reasoning; rather, it focuses on T2I retrieval and using the retrieval results to benefit T2I generation. To evaluate it, We also provide the retrieval-augmented compositional synthesis results in WikiArt and CUB datasets in Table 2. The results show that our method consistently improves generation quality compared with other baselines and MLLM backbones (gpt-image-1, gemini-2.0-flash) w/o RAG across all datasets, demonstrating that the retrieved sub-dimensional evidence is effectively leveraged during generation.
>
> We additionally add an experiment to test our Cross-modal RAG framework on a weaker generator, LaVIT. As shown in the Table, when we integrate LaVIT with our framework, the performance consistently improves across all three datasets and all metrics. These substantial gains indicate that the observed improvements result from the effectiveness of our proposed retrieval framework.
>
> | Method        |        |        |    WikiArt        |               |           |     CUB    |           |          |     ImageNet-LT    |           |
> |---------------|:---------------:|:--------:|:-----------:|:----------------:|:---------------:|:--------:|:-----------|----------------------:|:--------:|:-------:|
> |                      | CLIP↑        | DINO↑| SigLIP↑ | Style Loss↓&nbsp;&nbsp; | CLIP↑       | DINO↑ | SigLIP↑&nbsp;&nbsp;&nbsp;&nbsp; | CLIP↑ | DINO↑ | SigLIP↑ |
> |LaVIT | 0.689 | 0.485 | 0.721 | 0.036 | 0.676 | 0.245 | 0.647 | 0.662 | 0.365 | 0.652 |
> |LaVIT + our retrieval | 0.723 | 0.540 | 0.730 | 0.024 | 0.724 | 0.572 | 0.720 | 0.810 | 0.668 | 0.802 |

---

> > ### Author Response · Authors · 2025-11-22
> > **Official Comment by Authors (cont'd)**
> >
> > > W3: The design of Stage 1 and Stage 2 is very complicated but only performs the function of query decomposition, which has been widely investigated and applied in previous works. This paper lacks comparison with naive query decomposition methods, such as leveraging an LLM as the query decomposer.
> >
> > A3: We respectfully disagree with the claim that Stage 1 and Stage 2 “only perform the function of query decomposition.” In fact, the dual sub-dimensional decomposition of both queries and images is one of the core novelties, as also acknowledged by Reviewer ooGi
> > “the dual decomposition of queries and images into sub-dimensions is creative and distinguishes it from prior work.” Prior works only parse the text query into subqueries. In contrast, our approach goes beyond only textual decomposition. We also decompose images into the corresponding visual subembeddings with respect to the subqueries. This allows us to perform dense retrieval by calculating the similarity between the textual subquery embeddings and the corresponding visual subembeddings, rather than relying on coarse global image-text matching. Furthermore, we introduce a multi-objective joint retrieval mechanism to optimize the trade-off between sparse and dense retrieval results, selecting a Pareto-optimal set of images that maximizes subquery coverage. This dual decomposition and joint optimization framework has not been explored in previous RAG works and is essential for handling the complex or fine-grained queries addressed in our paper.
> >
> > Our method indeed uses an LLM as the query decomposer. If the concern is the comparison with non-LLM decomposer, we also implemented a rule-based decomposer using standard NLP techniques (Spacy-based Noun Phrase Chunking). While rule-based methods can identify basic noun phrases, we find that they struggle to extract clean "atomic entities"(a noun plus its modifiers) as defined in Appendix A. They often mishandle complex attribute associations that LLMs decompose correctly. Moreover, rule-based methods frequently fail to exclude non-visual/meaningless text (e.g., retaining "a photo of" or "an image of"), which introduces noise into the retrieval process. Therefore, we find that using an LLM as the decomposer leads to more accurate decompositions that better align with our task requirements.

---

> > > ### Comment · Reviewer_fk4a · 2025-11-25
> > >
> > > Thanks for the explanation; now that my concerns have been addressed, I've updated my score.

---

> > > > ### Author Response · Authors · 2025-11-26
> > > >
> > > > Thank you for your valuable feedback! We appreciate your support and are glad that our responses helped address your concerns.

---

### Official Review · Reviewer_MkFL · 2025-10-31

**Soundness:** 3
**Presentation:** 3
**Contribution:** 3
**Rating:** 4
**Confidence:** 4

**Summary:**

This paper proposes Cross-modal RAG, a retrieval-augmented framework for text-to-image generation. The core idea is to decompose a textual query into multiple sub-queries and retrieve corresponding image fragments in a cross-modal space through a hybrid sparse–dense retrieval mechanism. Subsequently, a multimodal large language model (MLLM) conditionally fuses these retrieved images to produce a composite visual output. The authors evaluate the approach on MS-COCO, Flickr30K, WikiArt, and ImageNet-LT, showing that it outperforms existing methods such as BLIP-2, SigLIP, and UniRAG in both fine-grained retrieval and image generation performance.

**Strengths:**

1. Proposes a sub-dimensional decomposition mechanism for cross-modal retrieval-augmented generation that is simple yet effective.
2. Achieves consistent improvements in both retrieval and generation.
3. The hybrid retrieval design (sparse + dense) achieves a good balance between accuracy and efficiency.

**Weaknesses:**

1. The paper’s novelty is not sufficiently articulated. The concept of Cross-modal RAG has appeared in prior studies (e.g., VisRet), so the authors should more clearly highlight their unique contribution—particularly the multi-dimensional decomposition mechanism for complex multimodal semantics. Comparative or visualization-based analyses (e.g., subquery–feature alignment) would help strengthen the differentiation.
2. The proposed “Pareto-optimal hybrid retrieval” remains largely heuristic and lacks theoretical analysis. The paper does not explain how the strategy guarantees or approximates Pareto optimality under given hyperparameter settings. The authors are encouraged to include sensitivity analyses of α and β or provide theoretical justification to demonstrate optimization stability and soundness.
3. The efficiency evaluation focuses mainly on parameter size but ignores the additional computational cost from multiple subquery retrievals and multi-image generation.
To ensure a fair comparison, the authors should additionally report:
• inference time and GPU memory usage under the same number of retrievals;
• baseline performance and latency when multi-retrieval is also applied.

**Questions:**

1. Have you analyzed the impact of α and β parameters on the balance between retrieval and generation?
2. How does the inference latency under multi-retrieval conditions compare with BLIP-2 or SigLIP?

---

> ### Author Response · Authors · 2025-11-22
>
> We’d like to thank the reviewer for acknowledging the effectiveness of our proposed approach. We address your raised points as follows.
>
> > W1: The paper’s novelty is not sufficiently articulated. The concept of Cross-modal RAG has appeared in prior studies (e.g., VisRet), so the authors should more clearly highlight their unique contribution—particularly the multi-dimensional decomposition mechanism for complex multimodal semantics. Comparative or visualization-based analyses would help strengthen the differentiation.
>
> A1: We appreciate the reviewer's comment. We would like to clarify that while the concept of cross-modal RAG has appeared in prior studies, our method introduces a fundamentally different formulation and technical pipeline, centered on the dual decomposition of text queries and images and a multi-objective joint retrieval framework, neither of which has been explored in previous T2I RAG works.
>
> First, unlike VisRet and other T2I retrieval methods that operate on global embeddings, our framework decomposes both the query and the candidate images into sub-dimensions. In detail, we decompose query into subqueries and images into the corresponding visual subembeddings with respect to the subqueries. This allows us to perform dense retrieval by calculating the similarity between the textual subquery embeddings and the corresponding visual subembeddings, rather than relying on coarse global image-text matching.
>
> Second, we formalize retrieval as a multi-objective optimization problem and introduce a multi-objective joint retrieval mechanism to optimize the trade-off between sparse and dense retrieval results, selecting a Pareto-optimal set of images that collectively cover all subqueries with no redundancy. Prior methods retrieve images based on a single global embedding thus cannot guarantee the full subquery coverage.
>
> Third, our generation is subquery-aware, where each retrieved image is associated with the subqueries it satisfies, enabling the MLLM to condition on only the relevant features from each image and discard the irrelevant features that prior methods usually preserve.
>
> Moreover, to further clarify these distinctions, we have a visualization comparison of retrieval and generation between our method and prior work in Figure 1 in the introduction section.
>
> > W2: Explain how the strategy guarantees or approximates Pareto optimality under given hyperparameter settings.
>
> Our paper clarifies that the hybrid retrieval is not heuristic: Theorem 3.2 proves the optimality of Algorithm 1. The weight vector $\alpha$ will enumerate all possible extreme values to ensure that every possible combination of subqueries is taken into account. And for any $0<\beta<\beta_{max}=\frac{\delta_{min}}{C_{\max}}$, the method returns all Pareto-optimal solutions to Eq.6. The complete proof is provided in Appendix C. More hyperparameter analysis of $\alpha$ and $\beta$ can be found in Appendix G.
>
> > W3: The additional computational cost when multi-retrieval is applied.
>
> A3:  We would like to clarify that the numbers reported in Table 3 are averaged over the entire MS-COCO 5K test set, and therefore already reflect the inference time and GPU memory usage under the same number of retrievals (i.e., 5K) for all methods. Also, the comparison in Table 3 is already performed in the same multi-retrieval setting to ensure fairness.
>
> > Q1: Have you analyzed the impact of $\alpha$ and $\beta$ parameters on the balance between retrieval and generation?
>
> A4: The parameters of $\alpha$ and $\beta$ only affect the retrieval objective in Eq.6, and influence generation indirectly through which images are retrieved. As Theorem 3.2 specifies, $\alpha$ enumerates all possible extreme values, and for any $0<\beta<\beta_{max}=\frac{\delta_{min}}{C_{\max}}$, the method is guaranteed to return all Pareto-optimal images. Therefore, when $\beta$ lies within this valid range, the retrieved images are Pareto-optimal, and our method is robust with respect to both retrieval and generation results.
>
> > Q2: How does the inference latency under multi-retrieval conditions compare with BLIP-2 or SigLIP?
>
> A5: Thank you for the question. We test the inference latency of BLIP-2 and SigLIP under the same setting. BLIP-2 (ViT-L) takes 21.95 ms/image and SigLIP takes 11.05 ms/image. Both are higher than CLIP (ViT-L).

---

> > ### Author Response · Authors · 2025-11-26
> >
> > Thank you for your time and effort in reviewing our paper! We noticed that the other reviewers have reached a consensus in the rebuttal, but we still have not received your response. As the rebuttal discussion phase is approaching its conclusion, we want to kindly follow up in case there are any further questions or clarifications you would like us to address.
> >
> > We have made extensive clarifications and provided additional experimental results as you suggested. We hope we have effectively addressed your concerns and clarified any potential misunderstandings, and we are looking forward to your feedback!

---

### Author Response · Authors · 2025-11-29
**Author Final Remarks**

We sincerely thank all the reviewers for their valuable review and follow-up responses. We also deeply appreciate the AC’s time in reviewing this rebuttal, especially given the substantial additional workload for this year's ICLR.

During the rebuttal period, we **responded to and resolved all of the reviewers’ concerns**. **Before the data leak on November 27**, our submission’s scores had raised from 6/4/4/4 to **6/6/6/4**. **Two reviewers explicitly stated that their concerns had been addressed and raised their scores**: reviewer jqS4 wrote “The rebuttal addresses most of my concerns. I have raised my score accordingly.” (5 days before the data leak), and reviewer fk4a wrote “Now that my concerns have been addressed, I've updated my score” (2 days before the data leak). The other two reviewers did not have time to respond before the discussion process was cut short. For reviewer MkFL, the concerns were mainly clarification questions. We carefully clarified them while pointing to the corresponding explanations in the manuscript. Reviewer ooGi only raised a few minor questions, which we also clarified and supported with additional experiments to validate the effectiveness of our framework.

**Summary of Contributions** We appreciate the reviewers for acknowledging our work:
* The **dual decomposition of queries and images into sub-dimensions** (ooGi: "creative and distinguishes it from prior work”; MkFL: “simple yet effective”; jqS4: "Cross-modal RAG addresses a core limitation of existing text-to-image RAG systems", ”sub-dimensional embeddings without explicit region cropping and aligns them to subqueries, which is both efficient and robust to noisy region proposals”).
* The **multi-objective Pareto-optimal formulation** is conceptually elegant and differs from common top-k retrieval (fk4a, jqS4).
* The **hybrid retrieval strategy** is technically sound and balances accuracy with efficiency (MkFL, fk4a, ooGi).
* The **subquery-aware generation** to condition on the satisfied subqueries (MKFL: “consistent improvements in both retrieval and generation”; fk4a:”conceptually elegant and potentially generalizable”; jqS4:”pragmatic and model-agnostic way to control conditioning at inference”)

We hope this summary assists the AC in evaluating our submission, and thank you again for your time and efforts in carefully considering our paper. If any further information or clarification would be helpful, we are more than happy to provide it.

Best regards,

Authors

---

### Meta-Review · Area_Chair_CjDm · 2026-01-01

**Summary:**

The reviewers broadly agree that the paper presents a coherent and potentially useful framework for text-to-image RAG that decomposes queries and images into sub-dimensions, retrieves a complementary set of images, and uses subquery-aware conditioning during generation. Reviewers also acknowledge strong empirical results on standard retrieval datasets and several generation benchmarks.

However, the main concerns affecting the decision are: (i) novelty/positioning relative to closely related “cross-modal RAG” and fine-grained retrieval work is not convincingly differentiated, and key comparisons are missing; (ii) core components rely on query decomposition and sparse caption matching, raising robustness concerns outside curated captioned datasets; (iii) the multi-objective / “Pareto-optimal” retrieval is not fully justified in a way that instills confidence for broad adoption; and (iv) the evaluation leaves uncertainty about generalization and real-world efficiency/latency trade-offs when multiple subqueries and multiple in-context images are used.

Given the remaining gaps, and despite a rebuttal that addresses several factual/clarification issues and adds some supportive ablations, the AC finds the overall evidence not yet strong enough for acceptance.

**Reviewer Concerns:**

Concerns addressed:
- Baseline correctness concern (fk4a) about BLIP-2 numbers appears to be clarified by pointing to cited results rather than a faulty reimplementation.
- Ablations for generation (jqS4): the rebuttal adds w/o-RAG and partial ablations (e.g., w/o Stage 1, w/o Stage 2) on generation benchmarks, partially addressing the “missing ablation” request.
- Weaker-generator sanity check (ooGi): the rebuttal includes an additional experiment integrating a weaker generator (LaVIT), which helps support that gains are not solely from a strong backbone.
- Some efficiency clarifications (MkFL): the rebuttal states that reported time/memory numbers are averaged over the full test set under the same retrieval setting, and provides latency figures for BLIP-2/SigLIP.

Outstanding concerns:
- Novelty/positioning remains weakly substantiated (MkFL): while the rebuttal describes differences conceptually, it does not fully resolve the reviewer’s request for stronger differentiation (e.g., clearer comparative analysis/visualization and sharper framing vs. prior “Cross-modal RAG”/VisRet-like lines).
- Missing key baselines/comparisons (ooGi): the lack of a direct comparison to FineRAG remains unresolved in a satisfying way (the rebuttal argues reproducibility issues, but this still leaves an evidence gap at decision time).
- Robustness to weak/noisy captions and lexical filtering (jqS4): the central concern that the sparse stage depends on dataset captions and may discard relevant images due to synonyms/paraphrases is not empirically resolved (the rebuttal suggests potential query/caption expansion but does not demonstrate robustness under realistic caption corruption/absence).
- Dependence on decomposition quality (ooGi, jqS4): while the rebuttal reports high “parsing accuracy” on specific benchmarks, this metric (string match / typo correction) does not substitute for demonstrating semantic decomposition quality across domains and failure modes (e.g., entity-attribute binding errors) and their downstream impact.
- Pareto/multi-objective retrieval justification (MkFL, jqS4): the rebuttal references theoretical results and appendices, but the practical question of stability/sensitivity and whether the approach is “principled” rather than parameter- and implementation-dependent remains insufficiently validated through targeted sensitivity experiments and real-world retrieval conditions.
- End-to-end efficiency and scaling (MkFL, jqS4): key questions remain about how latency/cost scales with the number of subqueries and in-context images, and whether there are diminishing returns; the rebuttal gives qualitative assertions and limited measurements rather than a comprehensive profiling study.
- Architectural complexity and brittleness: the proposed pipeline introduces several interacting components (LLM decomposition, sparse filtering, dense retrieval, Pareto-set selection, and subquery-aware in-context conditioning). Even if each step is individually reasonable, the composition increases engineering burden, tuning surface area, and potential failure modes (e.g., error propagation from decomposition to retrieval to generation). The rebuttal does not yet provide enough evidence (e.g., component necessity across datasets, sensitivity to hyperparameters, or simplified variants with comparable performance) to justify this added complexity for practical adoption.

**Reviewer Scores:**

- MkFL (hypothetical final rating: 4).
The reviewer’s core concerns are about novelty articulation, the principled nature of the Pareto-optimal hybrid retrieval, end-to-end efficiency scaling, and now the cumulative architectural complexity of the multi-stage pipeline. While the rebuttal provides clarifications (including references to theoretical claims and some latency discussion), the remaining gaps are largely about positioning, robustness, and convincing end-to-end evidence that the complexity is warranted. As a result, this reviewer would plausibly maintain a below-threshold score.
- fk4a (hypothetical final rating: 5). The reviewer indicated that their concerns were addressed and that they updated their score. Several of the original weaknesses (e.g., BLIP-2 number discrepancy and the “stage complexity” misunderstanding) appear to have been clarified. Nonetheless, given the remaining broader issues raised by other reviewers — especially robustness beyond captioned benchmarks, missing critical comparative baselines, and limited justification that the full multi-stage design is necessary — the updated score would likely settle at a borderline-but-still-below-threshold position rather than a confident accept.
- ooGi (hypothetical final rating: 5). The reviewer was initially slightly positive but explicitly noted missing direct comparison to FineRAG and reliance on LLM-based decomposition, plus a question about whether gains come from strong generators. The rebuttal adds a weaker-generator experiment (helpful) and discusses decomposer alternatives, but the key empirical comparison gap (FineRAG) remains, and the overall complexity of the pipeline raises further concerns about brittleness and deployability absent stronger comparative and robustness evidence. This reviewer would plausibly land on a borderline reject.
- jqS4 (hypothetical final rating: 5). The reviewer stated that most concerns were addressed and that they raised their score. The rebuttal adds requested ablations on generation benchmarks and provides some discussion around decomposition and Pareto set size. However, the most decision-relevant concern — robustness of caption-dependent sparse filtering and decomposition under noisy/absent captions and paraphrases — remains largely untested, and the multi-stage architectural complexity compounds this risk through error propagation. Therefore, while the score would likely increase from the initial review, a cautious borderline-below-threshold rating remains plausible.

---

### Decision · Program_Chairs · 2026-01-26

Reject